# TempO-seq and RNA-seq gene expression levels are highly correlated for most genes: A comparison using 39 human cell lines

Laura J. Word ●*, Clinton M. Willis, Richard S. Judson, Logan J. Everett, Sarah E. Davidson-Fritz, Derik E. Haggard, Bryant A. Chambers, Jesse D. Rogers ●, Joseph L. Bundy, Imran Shah, Nisha S. Sipes, Joshua A. Harrill ●*

Center for Computational Toxicology and Exposure, Office of Research and Development, United States Environmental Protection Agency, Research Triangle Park, North Carolina, United States of America

* drlauraword@gmail.com (LJW); harrill.joshua@epa.gov (JAH)

## Abstract

Recent advances in transcriptomics technologies allow for whole transcriptome gene expression profiling using targeted sequencing techniques, which is becoming increasingly popular due to logistical ease of data acquisition and analysis. As data from these targeted sequencing platforms accumulates, it is important to evaluate their similarity to traditional whole transcriptome RNA-seq. Thus, we evaluated the comparability of TempO-seq data from cell lysates to traditional RNA-Seq from purified RNA using baseline gene expression profiles. First, two TempO-seq data sets that were generated several months apart at different read depths were compared for six human cell lines. The average Pearson correlation was 0.93 (95% CI: 0.90–0.96) and principal component analysis (PCA) showed that these two TempO-seq data sets were highly reproducible and could be combined. Next, TempO-seq data was compared to RNA-Seq data for 39 human cell lines. The $\log_2$ normalized expression data for 19,290 genes within both platforms were well correlated between TempO-seq and RNA-seq (Pearson correlation 0.77, 95% CI: 0.76–0.78), and the majority of genes (15,480 genes, 80%) had concordant gene expression levels. PCA showed a platform divergence, but this was readily resolved by calculating relative $\log_2$ expression (RLE) of genes compared to the average expression across cell lines in each platform. Application of gene ontology analysis revealed that ontologies associated with histone and ribosomal functions were enriched for the 20% of genes with non-concordant expression levels (3,810 genes). On the other hand, gene ontologies annotated to cellular structure functions were enriched for genes with concordant expression levels between the platforms. In conclusion, we found TempO-seq baseline expression data to be reproducible at different read depths and found TempO-seq RLE data from lysed cells to be comparable to RNA-seq RLE data from purified RNA across 39 cell lines, even though the datasets were generated by different laboratories using different cell stocks.

---

**Data availability statement:** The datasets generated and analyzed during the current study are provided in Supplemental S1 File. The RNA-seq data is Protein Atlas Version 23 from the Human Protein Atlas website (https://www.proteinatlas.org/about/download, "RNA HPA cell line gene data" released 2023.06.19). All FASTQ files and aligned counts for the U.S. EPA TempO-seq data have been deposited into NCBI Gene Expression Omnibus under the accession number GSE288929 and are publicly available at: https://www.ncbi.nlm.nih.gov/geo/query/acc.cgi?acc=GSE288929. The R code is available through FigShare at: https://doi.org/10.23645/epacomptox.27341970.v1.

**Funding:** Partial funding for this research was provided by the US EPA through its Office of Research and Development under the Chemical Safety for Sustainability Strategic Research Action Plan FY2023-FY2026 (CSS.401.1.6) and through a cooperative research and development agreement between the U.S. EPA and Unilever (amendment 838-A018), which were both received by Joshua A. Harrill [JAH] at the U.S. EPA. Additional funding was provided through a federal Tox21 consortium collaboration between the US EPA and the Division of Translational Toxicology (DTT) at the National Institute for Environmental Health Sciences (NIEHS), which was received by Nisha S. Sipes [NSS] at the U.S. EPA (IAD 9587101-1).

**Competing interests:** The authors have declared that no competing interests exist. This manuscript has been reviewed by the Center for Computational Toxicology and Exposure, Office of Research and Development, U.S. Environmental Protection Agency and approved for publication. Approval does not signify that the contents reflect the views of the Agency, nor does mention of trade names or commercial products constitute endorsement or recommendation for use.

## Introduction

For many years, traditional non-targeted RNA-seq (referred to henceforth as RNA-seq) has been the accepted gold-standard technological platform used for transcriptomics research [1]. RNA-seq has been shown to be reproducible [2], but the sample preparation and data alignment processes are very involved. RNA-seq involves fragmentation of RNA, reverse transcription, amplification, and subsequent sequencing and alignment of reads to a reference transcriptome (which requires significant computing resources) [3]. There are different library preparation methods for RNA-seq, but a common approach enriches for messenger RNA (mRNA) through a poly-adenylated (poly-A) tail pull-down step [3], which is labor intensive and time-consuming. More recently, however, TempO-seq (Templated Oligo assay with Sequencing readout) has emerged as an alternative technology to RNA-seq [4]. TempO-seq is a gene expression profiling platform that uses templated detector oligos (DOs) to facilitate amplification of specific sequences. TempO-seq has been shown to be highly reproducible [5], and it has technical advantages of being compatible with cell lysates, thus saving time by eliminating the need for RNA purification and reverse transcription [4]. TempO-seq also has a simpler and more straightforward alignment process of sequencing data to the TempO-seq probe manifest. Since there has been an increase in TempO-seq data that is being produced and published in the scientific literature [6–15], there is a need to determine whether TempO-seq and RNA-seq data are comparable and/or are compatible enough to be combined for integrated analyses.

There are pros and cons to using TempO-seq compared to RNA-seq. The use of DOs means that TempO-seq will only detect mRNA for which there is a DO pair, which can either be an advantage or a disadvantage depending on the study goals. A potential disadvantage is that TempO-seq is not able to detect novel mRNA and may not generate much information about transcript variants, depending on the probes used. However, an advantage of TempO-seq's use of DOs is that there is typically only one DO per gene, which makes assignment of sequenced reads to gene counts relatively simple. Conversely, RNA-seq requires assigning multiple and often overlapping reverse transcribed, cDNA fragments to each gene, which makes assignment of sequenced reads to gene counts and other data analysis steps more complicated and variable [3]. An added benefit of TempO-seq's use of DOs is that non-functional competitor DOs (i.e., DOs without amplification sequences) can be used to attenuate the measurement of genes with very high expression levels. This use of non-functional competitor DOs is beneficial because sequencing reads are diverted to less abundantly expressed genes without the added expense of increasing read depth. Furthermore, because genes are attenuated at known ratios, native expression levels can be calculated through multiplication by a gene specific attenuation factor. Achieving a comparable attenuation effect with RNA-seq cannot be accomplished without additional steps during library preparation [16]. Thus, TempO-seq is cost effective at quantifying mRNA when analyzing many samples in bulk, such as in the case of large scale, high-throughput transcriptomics (HTTr) screening efforts like those at the US Environmental Protection Agency Office of Research and Development [7,13–15].

Limited case studies have demonstrated that TempO-seq is as consistent and sensitive at detecting changes in gene expression as RNA-seq. A study by Yeakley and colleagues found that TempO-Seq had a 99.6% specificity for DO binding to targeted mRNAs, was replicable across different read depths, had the ability to detect fold differences as low as 1.2-fold, and demonstrated high correlation with fold differences measured by RNA-seq ($R^2 = 0.9$) for more than 20,000 targets following exposure of MCF-7 cells to the histone deacetylase inhibitor Trichostatin A (TSA) [4]. The specificity and sensitivity of TempO-seq enabled creation of a reliable gene expression signature, which included some novel genes not previously linked to TSA exposure, and the signature was consistent across multiple human cell lines.

In another study, Bushel and colleagues compared data from the TempO-seq S1500+ surrogate transcriptome, which detects expression of 2,284 genes (approximately one tenth of the full transcriptome), to whole transcriptome RNA-seq [17]. Their analysis used purified RNA from liver samples of rats exposed to 15 chemicals covering 5 different mechanisms of action (MOAs) and showed some technological platform differences. TempO-seq data had a higher (better) signal to noise ratio, less unexplained variance, and more reproducibility between biological replicates compared to RNA-seq, which they found to be partly due to TempO-seq having less variation in detection of lowly expressed genes. Despite this platform difference, the interpretation of the chemical bioactivity data was concordant. A reasonable PCA grouping of transcriptomics profiles according to the different chemicals' MOAs was observed using both the TempO-seq and RNA-seq gene expression data [17], helping to validate the use of TempO-seq data in mechanistic toxicology studies.

Several studies have also compared the performance of TempO-seq and RNA-seq on fresh frozen and formalin-fixed paraffin-embedded (FFPE) samples. Turnbull and colleagues found that the two technologies produced comparable results for primary human breast cancer samples, particularly after correction for batch effects, and recommended TempO-seq as the preferable choice when analyzing samples with very limited quantity [18]. More recently, Cannizzo and colleagues determined that TempO-seq provided more consistent fold-change results for differentially expressed genes (DEGs) within frozen and FFPE mouse liver samples whereas RNA-seq had much greater variation between the two sample types [10]. Thus, they concluded that TempO-seq is more suited than RNA-seq for analyzing aged archival samples.

While these previous studies provide valuable data on TempO-seq performance relative to RNA-seq, a major limitation is that they only evaluated TempO-seq using purified RNA [4,10,17,18]. They did not investigate TempO-seq using tissue/cellular lysates and in the case of Bushel and colleagues, they only investigated the surrogate transcriptome. Moreover, Yeakley and colleagues investigated only a small number of human cell lines using whole transcriptome TempO-seq. Therefore, we propose to address these data gaps by comparing whole transcriptome TempO-seq data from cell lysates with RNA-seq from purified RNA while expanding analyses to evaluate baseline gene expression across 39 unique human cell lines using data from different laboratories. To our knowledge, this study compares the largest variety of different cell types between these two technologies to date and provides a methodology that researchers can use for determining if data from these two platforms can be combined for integrated analyses.

## Methods

### Coding

All data analysis was conducted using R version 4.4.1 [19].

### TempO-seq Phase 1 and Phase 2 data generation

All human cell lines in this study are commercially available and thus are exempt from needing ethics approval. The specimens were not collected specifically for our study and no one on the study team has access to the subject identifiers linked to the specimens, therefore the NIH does not consider this study to be human subjects research and therefore it does not require ethics approvals.

The human whole transcriptome TempO-seq assay (hWTv2) was used to generate gene expression profiles for 39 human-derived cell lines. The TempO-seq profiles for the 39 cell lines were generated as part of two interrelated studies conducted at the U.S. Environmental Protection Agency (US EPA), herein referred to as TempO-seq "Phase 1" and "Phase 2". Table 1 contains a complete list of cell lines with both TempO-seq and RNA-seq data, with a column that

**Table 1. List of 39 cell lines within the analyses.** The data comparison column specifies which analyses the cell line was included in: TvT is for the comparison between the two TempO-seq data sets (Phase 1 and Phase 2) and TvR for TempO-seq vs RNA-seq. Additional information about the cell culture conditions is available in the supplemental S1 File.

| Data comparison | TempO-seq phase | Cell line | Tissue origin | Disease or cell line | Growth mode |
|---|---|---|---|---|---|
| TvR | 1 | A549 | Lung | Carcinoma | Adherent |
| TvR | 1 | A704 | Kidney | Renal Cell Carcinoma | Adherent |
| TvR | 1 | ASC52Telo | Adipose Tissue | Mesenchymal Stem Cell | Adherent |
| TvR | 1 | BHY | Upper Aerodigestive Tract | Oral Squamous Cell carcinoma | Adherent |
| TvR | 2 | BT-483 | Breast | Ductal Carcinoma | Adherent |
| TvR | 2 | CAL-148 | Breast | Ductal Adenocarcinoma | Mixed |
| TvR | 2 | CAL-78 | Muscle | Chondrosarcoma | Adherent |
| TvT | 1, 2 | CCD-18Co | Colon | None (Fibroblast) | Adherent |
| TvT, TvR | 1, 2 | Daudi | Lymphoid | Burkitt's Lymphoma | Suspension |
| TvR | 1 | DMS 454 | Lung | Small Cell Lung Carcinoma | Adherent |
| TvR | 2 | DoHH2 | Lymphoid | B Cell Lymphoma | Suspension |
| TvR | 1 | DV-90 | Lung | Adenocarcinoma | Adherent |
| TvR | 2 | EFM-19 | Breast | Ductal Carcinoma | Adherent |
| TvR | 1 | HBEC3-KT | Lung | Bronchial Epithelia | Adherent |
| TvT, TvR | 1, 2 | HepG2 | Liver | Hepatoblastoma | Adherent |
| TvR | 2 | HOS | Bone | Osteosarcoma | Adherent |
| TvR | 2 | Hs.839.T | Skin | Melanoma | Adherent |
| TvR | 1 | hTERT-HME1 | Breast | Breast Epithelium | Adherent |
| TvR | 1 | hTERT-RPE1 | Eye | Pigmented Epithelium | Adherent |
| TvR | 2 | Huh-1 | Liver | Hepatoma | Adherent |
| TvR | 2 | Huh-7 | Liver | Hepatoblastoma | Adherent |
| TvR | 1 | HUVEC/TERT2 | Umbilical Cord | Vascular Endothelium | Adherent |
| TvR | 1 | KP-N-RT-BM-1 | Central Nervous System | Neuroblastoma | Adherent |
| TvT, TvR | 1, 2 | MCF7 | Breast | Adenocarcinoma | Adherent |
| TvR | 2 | MG-63 | Bone | Osteosarcoma | Adherent |
| TvR | 2 | MHH-CALL-4 | Lymphoid | B Cell Lymphoma | Suspension |
| TvT, TvR | 1, 2 | NCI-H1092 | Lung | Small cell lung cancer (stage E carcinoma) | Suspension |
| TvR | 2 | NCI-H1105 | Lung | Small Cell Lung Cancer | Suspension |
| TvR | 2 | NCI-H1436 | Lung | Small Cell Lung Cancer | Suspension |
| TvR | 2 | NCI-H2106 | Lung | Non-small Cell Lung Cancer | Suspension |
| TvR | 2 | NCI-H2171 | Lung | Small Cell Lung Cancer | Suspension |
| TvR | 2 | PLC/PRF/5 | Liver | Hepatoma | Adherent |
| TvR | 1 | RPTEC/TERT1 | Kidney | Proximal Tubule Epithelium | Adherent |
| TvR | 2 | SaOS-2 | Bone | Osteosarcoma | Adherent |
| TvR | 1 | SET-2 | Lymphoid | Acute Megakaryoblastic Leukemia | Suspension |
| TvR | 1 | SK-MEL-28 | Skin | Melanoma | Adherent |
| TvR | 2 | SU-DHL-6 | Lymphoid | Large/ B Cell Lymphoma | Suspension |
| TvR | 2 | T-47D | Breast | Ductal Carcinoma | Adherent |
| TvR | 1 | TIME | Skin | Dermal Microvascular Endothelium | Adherent |
| TvT, TvR | 1, 2 | U-2 OS | Bone | Osteosarcoma | Adherent |

denotes the TempO-seq data Phase (1, 2, or 1 and 2). The TempO-seq libraries for Phase 1 were sequenced to a target read depth of $6.0 \times 10^6$ mapped reads per sample using an unattenuated version of the hWTv2 assay. Due to a lack of *a priori* knowledge about baseline gene expression within all 39 different cell types at the time this study was conducted, we decided to use an

unattenuated version of the TempO-seq assay to ensure we captured the expression of all genes within each cell type to the extent possible. The TempO-seq libraries for Phase 2 were generated and sequenced approximately seven months later to a target read depth of $4.5 \times 10^6$ mapped reads per sample using an unattenuated version of the hWTv2 assay. All cell lines were cultured in the growth medium recommended by the vendors from which the cell lines were sourced, and Phase 1 and Phase 2 data were generated by the same laboratory technician and from the same cryostocks. Cell culture conditions and the procedure for generating TempO-seq lysates are described in detail in the supplemental S1 File.

Of the 39 cell lines included in our analysis, six of the cell lines were sampled in both TempO-seq Phase 1 and Phase 2, as listed in Table 1. These six cell lines were used to evaluate the feasibility of combining TempO-seq datasets with variable read depths prior to comparison of TempO-seq and RNA-seq gene expression profiles. Triplicate samples of the lysates from each cell line in each TempO-seq phase of experimentation were profiled and served as our technical replicates.

The TempO-seq data included 22,537 probes, in which some probes were for the same gene. When there were multiple TempO-seq probes for the same Ensembl gene identifier (ensemble_id), the maximum count value was taken for that gene for each cell line and technical replicate separately which resulted in a dataset with 19,703 unique genes. TempO-seq count data for each sample was then normalized to counts per million (CPM) using the equation:

$$CPM = \frac{Gene\ count\ *\ 10^6}{\sum \left( All\ gene\ counts\ for\ that\ sample \right)} \quad (1)$$

CPM for each replicate was then log transformed using the formula $\log_2(CPM+1)$. We add one to the CPM, such that when CPM is zero the resulting log-transformed value is zero rather than negative infinity.

## Comparing TempO-seq Phase 1 vs TempO-seq Phase 2 data

Pearson correlations were calculated for comparing gene counts between TempO-seq technical replicates within each Phase separately and between the averaged technical replicate data from TempO-seq Phases 1 and 2 in matching cell lines. Principal component analysis (PCA) was then performed to evaluate how well the data grouped together by cell line or by TempO-seq Phase (i.e., TempO-seq Phase 1 or Phase 2) using the base R *stats::prcomp* function. For this analysis, the gene expression data was centered but not scaled because the data sets use the same units. To understand the comparative influences of cell line and TempO-seq Phase on the PCA results, analysis of variance (ANOVA) was run on individual principal components (PCs) using the *aov* function within the base R *stats* package (ex. Distance_matrix_PC1 ~ Cell_line + Phase) and permutational multivariate analysis of variance (PERMANOVA) [20] was run on the full PCA distance matrix data across all PCs (Distance_matrix_all_PCs ~ Cell_line + Phase) [21], set to 999 permutations, using the *adonis2* function within the *vegan* R package [22]. The distance matrix was calculated using the base R function *dist* on the PC data to calculate the Euclidean distances for inputting into the PERMANOVA. When PERMANOVA was run, the function *betadisper* within the *vegan* package was used to test for dispersion effects and the corresponding principal coordinate analysis (PCoA) plots were generated to qualitatively visualize the relative magnitude of the impacts from location vs dispersion [23]. While PCA reduces data to a smaller number of key components that capture the most variation (PCs), PCoA is a technique that visualizes how different samples relate to one another based on their overall dissimilarity, often by using distance measures [23].

### RNA-seq data source

The RNA-seq data was previously generated by the Human Protein Atlas (HPA) project [24]. We downloaded Protein Atlas Version 23 from their website (https://www.proteinatlas.org/about/download, "RNA HPA cell line gene data" released 2023.06.19). Then we filtered it for the 39 cell lines that overlapped with our data to enable comparison between the downloaded HPA RNA-seq data and the TempO-seq data from the U.S. EPA (Table 1). A large amount of the data is from the Cancer Cell Line Encyclopedia (CCLE), which reports using oligo-dT beads to select polyadenylated mRNA [25]. If the HPA project used technical or biological replicates for any of the cell lines, then these were not reported in the data available for download. The HPA RNA-seq data used Ensembl version 109, and the expression values were reported as TPM, which were then re-normalized to one million reads per sample after non-coding transcripts were removed (denoted by HPA as pTPM).

### Preparing data for comparison of TempO-seq and RNA-seq

The two internal TempO-seq datasets, Phase 1 and Phase 2, were combined for comparison to the HPA RNA-seq data. For the HPA data, only one set of data is publicly available (no replicate data). Therefore, we averaged the TempO-seq CPM replicate data (for Phase 1, Phase 2, or both when available, see Table 1) on a per gene basis for comparison to the HPA RNA-seq data set.

The 39 overlapping cell lines evaluated are shown in Table 1. The HPA RNA-seq data was reported as transcripts per million (TPM), which quantifies the number of transcripts one would observe if one were to sequence one million full length transcripts while taking into account gene length. Accounting for gene length in RNA-seq normalization is important due to longer genes having a greater number of reads from fragments aligning to different sections of the longer gene. For TempO-seq, on the other hand, genes are quantified using DO pairs that produce reads specific to one location per gene for sequencing and is not so heavily influenced by gene length differences. Therefore, TempO-seq normalization does not require accounting for gene length. Thus, direct comparison between CPM and TPM was appropriate for our analyses. For simplicity, we will refer to CPM and TPM collectively as EPM, for expression per million.

Data was reduced to only include overlapping genes based on ensemble_id that were measured in the TempO-seq data and the RNA-seq data sets. After reducing the 19,703 TempO-seq gene set and the 20,162 RNA-seq gene set to the 19,290 overlapping genes that were detected within both platforms. We then re-scaled the remaining EPM values to a total of 1 million per sample and then $\log_2$ transformed the values for statistical analysis using the formula $\log_2(EPM+1)$. Note that a value of one was added to EPM so that when CPM or TPM is zero, $\log_2(EPM+1)$ returns a value of zero instead of returning negative infinity.

### Initial analysis of TempO-seq vs RNA-seq

Pearson correlations were calculated for comparing the TempO-seq data to the RNA-seq data and the correlations were averaged for matching cell lines. PCA was then performed to evaluate how well the data grouped together by cell line or by technology platform type (i.e., TempO-seq and RNA-seq). The gene expression data was centered but not scaled because the data sets use equivalent units. To understand the comparative influence of platform on the PCA results, ANOVA was run on individual principal components (PCs) (ex. Distance_matrix_PC1 ~ Platform) and PERMANOVA was run on the full PCA distance matrix data across all PCs (Distance_matrix_all_PCs ~ Platform), set to 999 permutations, along with evaluating dispersion effects using the *betadisper* function in R and generating PCoA plots. Cell line was not

included as a factor in the PERMANOVA analysis when comparing the TempO-seq vs RNA-seq PCs since there were only two data points per cell line (i.e., one point per cell line-platform pair), which is insufficient for making a reliable inference about cell line effects.

## Understanding the platform divergence: Non-concordance determination and pathway analysis

Non-concordant gene determination: We determined that genes with the largest differences in expression level between TempO-seq and RNA-seq were driving the platform divergence identified in the PCA. To identify which genes with non-concordant expression levels to remove, we aimed to subset the data until the variance explained by platform effects was resolved to a minimal level of influence of < 10% (evidenced by PERMANOVA platform $R^2$ < 0.10).

We started by calculating the absolute value of the difference in $\log_2(EPM + 1)$ between TempO-seq and RNA-seq and calculated the average interquartile range across all 39 cell lines for each of the 19,290 overlapping genes. Then, we repeated PCA with different cutoffs that corresponded to removing genes at the tails of the average relative $\log_2$ difference distribution for genes that met a minimum expression threshold (average TempO-seq expression ≥ 5 CPM and/or average RNA-seq expression ≥ 5 TPM) until the $R^2$ for platform effect showed that < 10% of the variance in the data was explained by platform type. For example, some of the cutoffs tested included removal of genes outside of the 10th and 90th percentiles, 11th and 89th percentiles, 12th and 88th percentiles, etc. Note that the minimal expression threshold requirement was added because the $\log_2$ values for genes with low values (< 5 EPM) contain mostly noise. For genes with < 5 CPM and < 5 TPM, TempO-seq and RNA-seq essentially agreed that those genes are not expressed. Thus, we categorized genes with < 5 EPM as effectively concordant.

This stepwise procedure resulted in the removal of 3,810 non-concordant genes with an average $[\log_2(EPM + 1)$ difference] <-2.09 or > 1.47 for TempO-seq minus RNA-seq, which were the average 13th and 87th percentile tails across all cell types for the boxplots in Fig 3, respectively (see S2 File, tab "TR_Nonconcordant genes" for a visualization of the nonconcordant genes removal bands). A heatmap was generated using the *ComplexHeatmap* R package [26] to show the comparative gene expression levels within TempO-seq and RNA-seq cell lines for the non-concordant genes.

Biological pathways ontology analysis: To better understand if any Gene Ontology (GO) signatures were more or less reproducible between the two technologies, we calculated odds ratios (OR) of how much more likely a set of GO signature genes was to be represented in the list of genes that were non-concordant between the TempO-seq and RNA-seq data sets (see Table 2). We subset genes to the 10,487 genes with CPM ≥ 5 and/or TPM ≥ 5 so that only genes that were detected by the technologies were included in this analysis. We evaluated 10,461 GO signatures from the Molecular Signatures Database Human Collections for release 2023.2 (MSigDB 2023.2.Hs) C5 collection on April 25th, 2024 [27–29]. MSigDB v2023.2 is based on gene annotation data from Ensembl Release 110, so RNA-seq and TempO-seq gene symbols were lifted from the older Ensembl Release versions to Ensembl Release 110 for this analysis.

We required at least 10 genes from the GO signature to be within our data for this analysis on the 10,487 expressed genes, which we based on the minimal number of genes required for transcriptomics gene signature analysis in previous research using whole transcriptome TempO-seq [7]. We also required at least 50% of the genes within the GO signature to be remaining within the 10,487 expressed genes that were retained for analysis. This resulted in 3,935 GO signatures being retained in our analysis, however, the ORs for the full list of 10,461 GO signatures is available in the supplemental S2 File within the "TR_All GO signatures"

**Table 2. Gene ontology (GO) odds ratio (OR) calculations.** This is the 2x2 table used for the OR calculations for the 10,487 expressed genes (CPM ≥ 5 and/or TPM ≥ 5). The equation used was OR = $(a/b)/(c/d)$ in which $a$ was the number of GO signature genes that were non-concordant, $b$ was the number of non-concordant genes that were not within the GO signature, $c$ was the number of concordant genes within the GO signatures, and $d$ was the number of concordant genes that were not within the GO signature. The sum of $a$ plus $b$ equaled the total number of non-concordant genes (3,810) and the sum of $c$ plus $d$ equaled the total number of concordant genes (6,677).

| | Within GO signature | Not within GO signature | Totals |
|---|---|---|---|
| Non-concordant genes | $a$ | $b$ | 3,810 genes |
| Concordant genes | $c$ | $d$ | 6,677 genes |
| | $(a+c)$ | $(b+d)$ | $(a+c)+(b+d) =$ 10,487 genes |

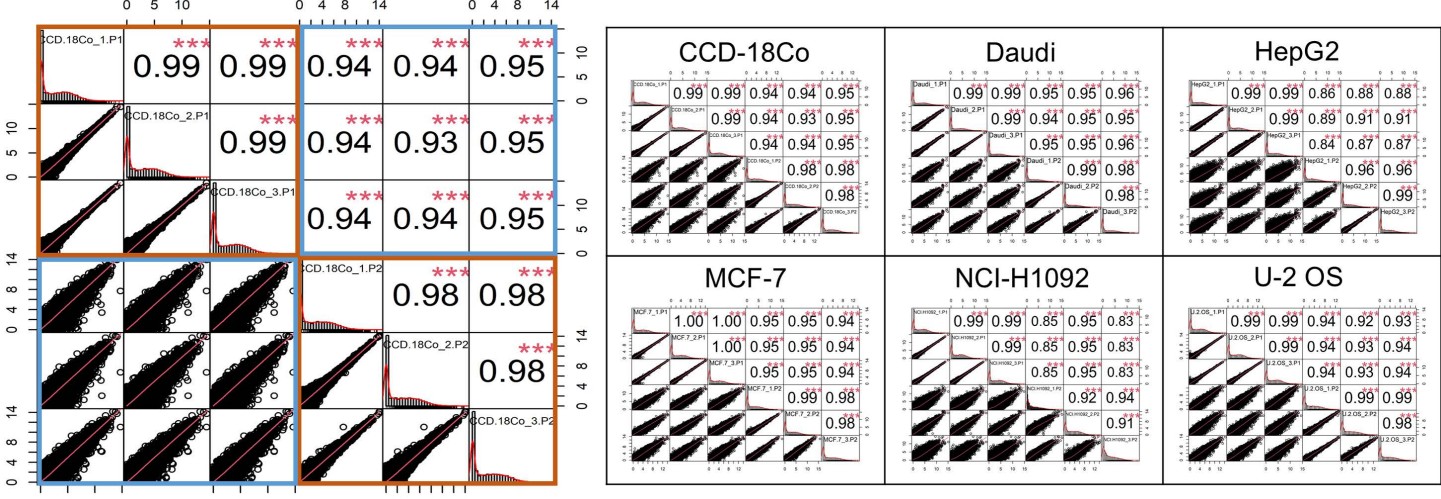

**Fig 1. Pearson correlations for the TempO-seq Phase 1 and Phase 2 technical replicates data.** Each cell line was analyzed separately. (A) Representative correlation plot for the CCD-18Co cell line. The cell line is indicated above each histogram in the center diagonal plots, followed by the technical replicate number (1 through 3), and then P1 or P2 for which TempO-seq data set phase the replicate was from (P1 is TempO-seq Phase 1 and P2 is TempO-seq Phase 2). The order within the top left orange box is: Phase 1, replicates 1, 2, 3, and the order within the bottom right orange box diagonal of histograms is: Phase 2 replicates 1, 2, 3. The top left and bottom right quadrants show Pearson Correlations between technical replicates in the same data set (orange boxes). Pearson Correlations between TempO-seq Phase 1 vs Phase 2 data sets are shown in the bottom left and top right quadrants (blue boxes). (B) Pearson correlations between TempO-seq Phase 1 and Phase 2 data sets for all six overlapping cell lines. Organization and annotation of these panels is the same as described in (A).

tab (to view only the 3,935 included GO signatures, 'sig', subset column D "sig_genes_inlists" to ≥10 and column F "percent_sig.genes_withinlists" to ≥50). Additionally, the p-values for the GO signatures were corrected for multiple comparisons using the false discovery rate (FDR) approach [30]. ORs above 1 indicate GO signatures were enriched with more non-concordant genes between TempO-seq and RNA-seq, and ORs below 1 indicate GO signatures that were enriched with more concordant genes between TempO-seq and RNA-seq.

## Resolving the platform divergence

The gene expression data was normalized by calculating the relative log expression (RLE) compared to the average expression level across all cell lines in each platform separately. We did this for each gene and cell line by dividing each gene's $\log_2(EPM + 1)$ by the average $\log_2(EPM + 1)$ across all cell lines for the TempO-seq data and the RNA-seq data separately for

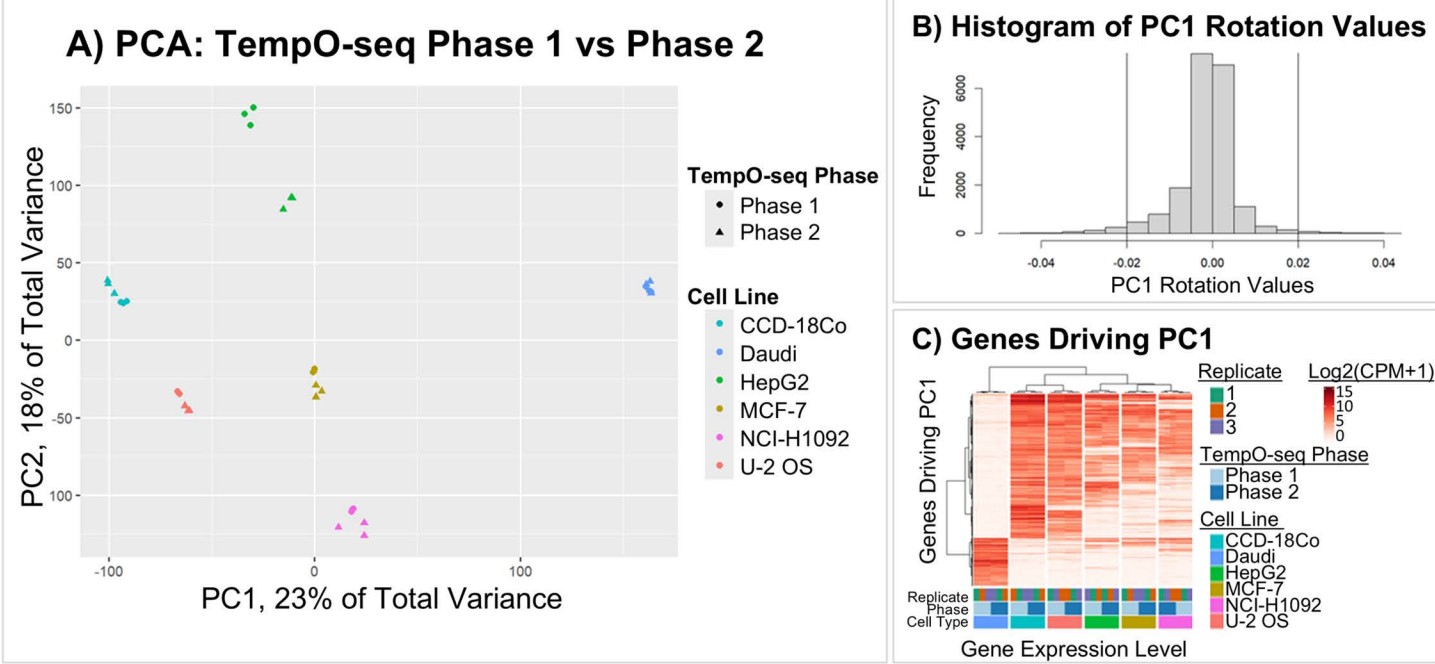

**Fig 2. TempO-seq Phase 1 vs TempO-seq Phase 2 are highly comparable.** (A) Principal component analysis (PCA) using data for all 19,703 genes within the two TempO-seq data sets, Phase 1 and Phase 2. These TempO-seq data sets were generated from the same cryostocks but different cultures that were spaced approximately seven months apart. Cell lines are different colors, and the shape indicates whether the data were from Phase 1 (circle) or Phase 2 (triangles) for each of the three technical replicates in the data sets. Two of the three technical replicates for HepG2 Phase 2 overlap each other on the plot. PCA showed that the data grouped well for all replicates within each data set and across both data sets, with the exception of a comparatively larger difference between the two data sets for HepG2 across principal component 2 (PC2). (B) This histogram shows the distribution of PC1 rotation scores for each gene analyzed in the PCA. (C) This heatmap shows that the expression of genes that drive PC1. Rows are genes and columns are each replicate for the cell lines. The intensity of red shading indicates $log_2$(CPM+1) expression level and column breaks were added for the dendrogram's six main branches, which happened to line up with the six different cell lines. Genes with the highest rotation values driving PC1 had consistent levels of expression across the technical replicates and both TempO-seq data sets, Phase 1 and Phase 2, within each of the cell lines.

all 19,290 genes. PCA, PERMANOVA, dispersion effects tests, and PCoA were then repeated using the RLE values and the Pearson correlations between TempO-seq and RNA-seq cell lines were re-run. For hypothetical "Gene X" the equation for RLE was:

$$RLE \text{ } for \text{ } Gene \text{ } X \text{ } = \text{ } log_2 \left( \frac{(EPM \text{ } + \text{ } 1) \text{ } for \text{ } gene \text{ } X \text{ } within \text{ } a \text{ } single \text{ } cell \text{ } line}{Average \text{ } (EPM \text{ } + \text{ } 1) \text{ } for \text{ } gene \text{ } X \text{ } across \text{ } all \text{ } 39 \text{ } cell \text{ } lines} \right) \quad (2)$$

## Results

### TempO-seq profiles are highly reproducible at different read depths

The two TempO-seq data sets (Phase 1 and Phase 2), which each contained three technical replicates profiled seven months apart for six overlapping cell lines at different read depths (6 million reads for Phase 1 and 4.5 million reads for Phase 2), were compared using Pearson correlations (see Fig 1). The average Pearson correlation of $log_2$(CPM+1) normalized gene counts across technical replicates was 0.99 (95% CI: 0.99–0.99) for Phase 1, 0.97 (95% CI: 0.96–0.98) for Phase 2, and was 0.98 (95% CI: 0.97–0.99) when averaged across both Phase 1 and Phase 2. When comparing the averaged technical replicate data across the two TempO-seq phases, the average Pearson correlation was 0.93 (95% CI: 0.90–0.96).

Next, PCA was conducted to evaluate grouping of samples using the gene expression data for the 19,703 genes within TempO-seq Phase 1 and Phase 2. Visual inspection of the PCA

showed that the two TempO-seq data sets grouped together by cell line for all six replicates along principal component 1 (PC1) and not by whether data was generated during Phase 1 or Phase 2 (see Fig 2A). ANOVA on PC1 showed that 99.6% of the variance was explained by cell line ($R^2$ = 0.996, p <2e-16) and only 0.09% of the variance within PC1 was due to the TempO-seq phase designation ($R^2$ = 9e-4, p = 4e-3). For PC2, PCA showed that the only cell line with more divergence between the two TempO-seq datasets was HepG2.

When evaluating the equation PCA_distance_matrix ~ Cell Line + TempO-seq Phase across all PCs using PERMANOVA, the cell line explained 78% of the variance ($R^2$ = 0.78, p = 0.001), and the TempO-seq phase accounted for less than 10% of the variance ($R^2$ = 0.093, p = 0.001). The *betadisper* function confirmed that this was due to data centroid location effects, not due to dispersion of the data (platform p-value was 0.6). Visual inspection of the PCoA for platform showed extensive overlap between groups, with only a small shift in the centroids; overall, the location and dispersion for platform effects were very similar for TempO-seq Phase 1 vs Phase 2 data. For cell line, the dispersion statistic on the triplicate data was significant (p = 0.001) and visual inspection of the PCoA plot showed that this may be because the HepG2 data had greater dispersion than the other 5 cell lines (see the supplemental S3 File).

To visualize the genes that were driving differences in PC1, we plotted PC1 rotation values (Fig 2B) and then visualized the expression of high rotation genes at the tails with absolute value > 0.02. The genes driving PC1 variance had differing gene expression patterns across the cell lines, but these genes had consistent expression levels for all six replicates within each cell line for both Phase 1 and Phase 2, as expected based on the PCA clustering pattern (see Fig 2C).

Based on these analyses, we concluded that any batch effects between TempO-seq Phase 1 and Phase 2 were minimal in comparison to the relative differences between cell lines. The Phase 1 and Phase 2 data were therefore combined for all subsequent comparisons to RNA-seq data.

## TempO-seq vs RNA-seq platform differences are consistent across cell lines

We calculated the relative log difference in $\log_2$(EPM + 1) values between TempO-seq and RNA-seq, for each gene across all 39 cell lines, and found these values to be consistently centered around zero with similar distributions and numbers of outliers (i.e., genes with non-concordant expression levels between technologies) across all cell lines for the 19,290 overlapping genes (see Fig 3). The average 50th percentile of TempO-seq minus RNA-seq relative log difference in $\log_2$(EPM + 1) was -0.04, which indicates that there was only a slight difference on average between TempO-seq and RNA-seq expression values. When assessing the interquartile ranges (IQR) for the TempO-seq minus RNA-seq relative log difference in $\log_2$(EPM + 1), the narrowest IQRs indicated that the smallest platform divergence was observed for Daudi, HUVEC/TERT2, and hTERT-HME1. Conversely, the widest IQRs were observed for NCI-H1436, NCI-H2106, and NCI-H1105, indicating the largest platform divergence was associated with these cell lines.

PCA of TempO-seq versus RNA-seq data showed a clear platform divergence pattern captured by PC1 for all 39 cell lines that were evaluated (see Fig 4). ANOVA revealed that the platform categorization explained 97% of the variance for PC1 ($R^2$ = 0.97, p <2e-16). Although, visual inspection of the PCA plots encouragingly shows that clustering by cell line was prominent within PC2 and PC3 (for PC3 see the supplemental S2 File).

PERMANOVA results across all PCs for TempO-seq vs RNA-seq $\log_2$(EPM+1) showed that, in total, the platform effect accounted for 31% of the total variance ($R^2$ = 0.31, p = 0.001). Dispersion effects testing on the distance matrix for all PCs showed significant effects for platform (p = 0.016). Since there is a significant result for dispersion effects, the PERMANOVA results need to be interpreted with the aid of the PCoA plots. Visual inspection of the PCoA

shows that, in addition to a relatively smaller difference in the dispersion, there is also a clear and pronounced difference in the centroid locations of the TempO-seq vs RNA-seq data without any overlap, confirming that there is a clear platform divergence within the $\log_2$(EPM+1) data (see the supplemental S3 File).

## Analysis of the non-concordant genes

Genes with the greatest difference in RLE levels between TempO-seq and RNA-seq were progressively removed until the PERMANOVA variance explained ($R^2$) for platform effect across all PCs was < 10% ($R^2$ = 0.099, p = 0.001, see the supplemental S2 File), as described in the methods. Of note, this PERMANOVA result on the $\log_2$(EPM+1) data after removal of non-concordant genes is reliable for data centroid location effects because the *betadisper* test for dispersion effects was not significant for platform (p = 0.07, see the supplemental S3 File). To better understand the expression patterns of the genes driving the platform divergence observed in the initial PCA, we visualized the individual expression levels of the 3,810 most non-concordant genes on a heatmap (see Fig 5). We found that the expression levels of most non-concordant genes were consistently higher or lower in TempO-seq relative to RNA-seq, regardless of cell line. Notable gene families included many ribosomal protein gene families (RPL, RPS, MRP, etc) that had lower expression in TempO-seq data than RNA-seq data, and many histone protein gene families (H2AC, H4C, etc) that had higher expression levels in the TempO-seq data.

We also calculated the average fold difference between RNA-seq and TempO-seq EPM across all data, which was 1.18 (95% CI: 1.18–1.19) for all genes and this decreased to 1.13 (95% CI: 1.12–1.13) when subset to concordant genes only, with RNA-seq normalized

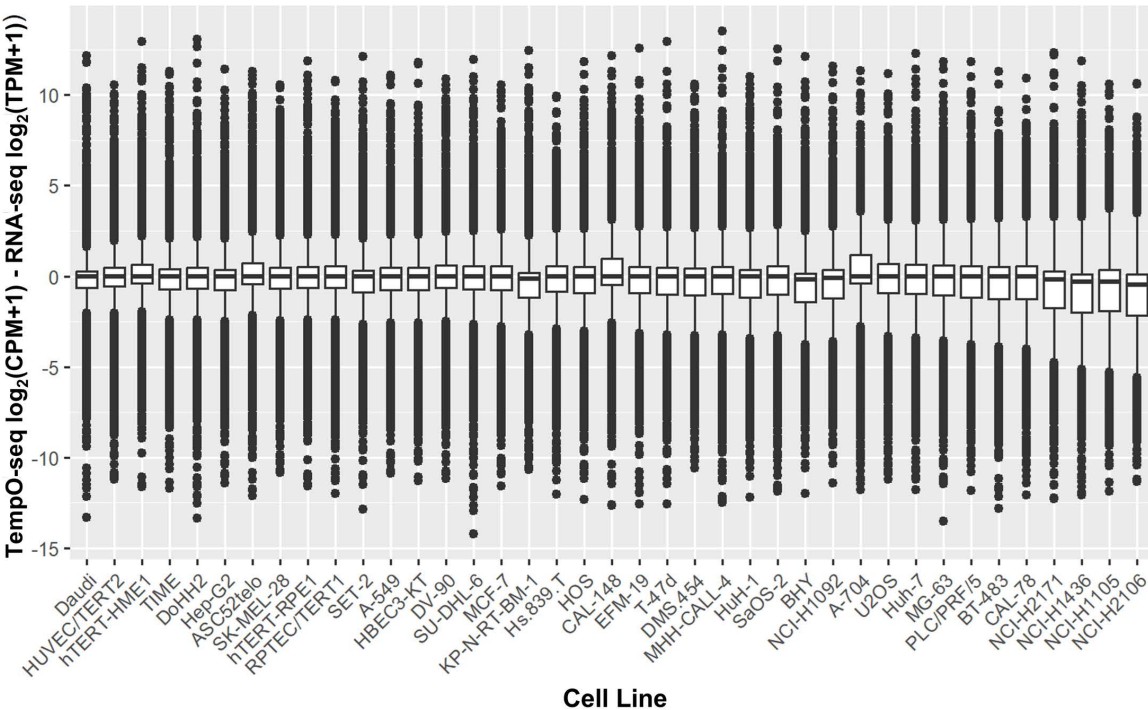

**Fig 3. Boxplots showing TempO-seq $\log_2$(CPM+1) minus RNA-seq $\log_2$(TPM+1) for 19,290 genes for all 39 cell lines.** The cell lines are arranged based on the size of their inter-quartile range, from smallest to largest.

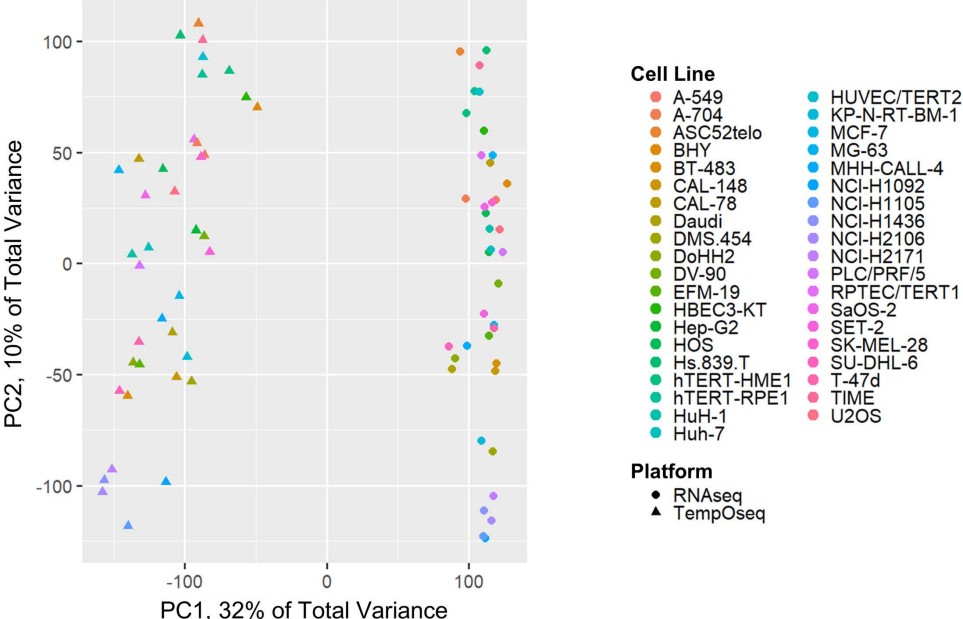

**Fig 4. Principal component analysis (PCA) plots for TempO-seq vs RNA-seq log₂ data.** This PCA is based on $\log_2$(expression per million + 1) (abbreviated to $\log_2$(EPM+1)) for 19,290 overlapping genes within TempO-seq and RNA-seq. Principal component 1 (PC1) explains nearly one third of the total variance and has a clear platform divergence for RNA-seq Human Protein Atlas data compared to TempO-seq Phase 1 and Phase 2 data.

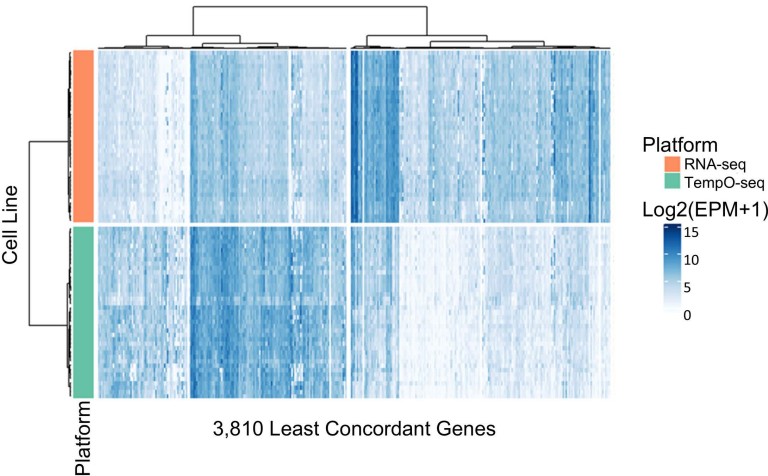

**Fig 5. Heatmap of the expression levels for the 3,810 least concordant genes.** These are the genes with the highest values for the relative log difference between TempO-seq and RNA-seq. The columns are genes and the rows are the cell lines analyzed, with the RNA-seq expression levels shown on the top half of the rows (orange band) and the TempO-seq expression levels shown on the bottom half of the rows (green band). The blue color shading within the heatmap indicates $\log_2$(expression per million + 1) (abbreviated to $\log_2$(EPM+1)) level for each gene; ranging from white for lowly expressed genes corresponding to 0 $\log_2$(EPM+1) to deep blue for highly expressed genes corresponding to 15 $\log_2$(EPM+1). Genes in the left branch of the dendrogram generally had higher expression in TempO-seq than RNA-seq, and genes in the right branch had lower expression in TempO-seq and higher expression in the RNA-seq data across all 39 cell lines.

TPM expression values trending higher relative to TempO-seq CPM values. Similarly, the average Pearson correlations were 0.77 (95% CI: 0.76–0.78) across all 19,290 overlapping genes and increased to 0.90 (95% CI: 0.88–0.91) when subset to the 15,480 concordant genes.

Next, an odds ratio (OR) analysis was performed to determine if the non-concordant genes were associated with known biological functions or molecular signaling pathways. For the 3,810 expressed genes with an average CPM ≥ 5 and/or TPM ≥ 5 that we determined to be non-concordant, we calculated ORs for enrichment of Gene Ontology (GO) signatures retrieved from MSigDB. Of the 10,461 GO signatures retrieved from MSigDB, 3,935 met our requirements: to be included a signature must have had at least 10 genes that were within the 10,487 overlapping TempO-seq and RNA-seq expressed genes (average CPM ≥ 5 and/or TPM ≥ 5) and those 10 or more genes had to represent at least 50% of the total genes within the signature.

We identified 27 GO signatures that had an FDR < 0.05 (see Table 3), of which 22 GO signatures had an OR > 1 (containing a higher proportion of non-concordant genes than concordant genes). The 22 more non-concordant GO signatures predominantly had functions associated with chromatin, chromosome centromeres, and cytosolic ribosomal subunits. All of the GO signatures with OR > 1 contained multiple histone family genes (H2AC, H2B, H4C, etc) and/or many ribosomal family genes (RPS, RPL, MRP, etc).

We found that 5 GO signatures had OR < 1 and FDR < 0.05 (containing a higher proportion of concordant genes than non-concordant genes). Table 3 shows calculations for 1/OR, which represents what the odds are for the GO signature to contain a greater proportion of concordant genes than non-concordant genes between the TempO-seq and RNA-seq data. The most concordant GO signatures include functions for the basement membrane, cellular structure, and kinase activity. The supplemental S2 File includes the full list of signatures, ORs, and genes within each GO signature.

## PC1 platform divergence was successfully resolved using RLE normalization

The platform divergence between TempO-seq and RNA-seq on PC1 was able to be resolved using RLE normalization of the 19,290 overlapping genes within both platforms (see Fig 6). ANOVA on PC1 RLE data showed that the platform accounted for none of the variance following this relative log expression normalization step ($R^2 = 0$, $p = 1$).

PERMANOVA on all PCs for RLE data also demonstrated that TempO-seq vs RNA-seq platform designation accounted for none of the variance ($R^2 = 0$, $p = 1$). Dispersion effects testing on the distance matrix for all PCs showed significant effects for platform ($p = 0.018$); visual inspection of the PCoA shows that there is near complete overlap of the TempO-seq and RNA-seq data, with very similar dispersions and centroid locations, indicating that any remaining platform differences after RLE are minimal (see the supplemental S3 File). Additionally, visual inspection of PC1, PC2, and PC3 show that the data grouped together well by cell line, which further supports the finding that the platform divergence was resolved by the RLE normalization approach.

Additionally, Pearson correlations were run both before and after RLE normalization (see Fig 7). Initially, the TempO-seq vs RNA-seq Pearson correlations between matching cell types averaged 0.77 (95% CI: 0.76–0.78) and the correlations between non-matching cell lines averaged 0.64 (95% CI: 0.64–0.65). After RLE normalization, Pearson correlations between matching cell types averaged 0.71 (95% CI: 0.67–0.74) and the correlations between non-matching cell lines averaged -0.02 (95% CI: -0.03 to -0.01).

**Table 3. Significant Gene Ontology (GO) signatures with false discovery rate (FDR) adjusted p-values < 0.05.** The transcriptomics dataset included a total of 10,487 genes that met a minimum expression threshold (≥5 CPM in TempO-seq or ≥5 TPM in RNA-seq), of which 3,810 genes were classified as non-concordant and 6,677 genes were classified as concordant. The GO analysis included 10,461 GO signatures, of which 3,935 passed our inclusion criteria for analysis to have at least half the signature's genes and a minimum of 10 genes within our list of 10,487 minimally expressed genes. GO signatures with odds ratios (ORs) > 1 had greater odds of non-concordant levels of expression between TempO-seq and RNA-seq for the genes within the signature. The 1/OR column shows the odds that expression of the GO signature genes are concordant between the TempO-seq and RNA-seq platforms (most reproducible) for the signatures with OR < 1 to improve interpretation.

| GO signature | Genes (n) | Genes in analysis | % genes in analysis | OR | 1/OR | FDR p-value |
|---|---|---|---|---|---|---|
| GOBP_PROTEIN_LOCALIZATION_TO_CENP_A_CONTAINING_CHROMATIN | 18 | 17 | 94 | 28.15 | – | 5.6E-04 |
| GOCC_CHROMOSOME_CENTROMERIC_CORE_DOMAIN | 19 | 18 | 95 | 14.07 | – | 3.1E-03 |
| GOMF_STRUCTURAL_CONSTITUENT_OF_CHROMATIN | 97 | 67 | 69 | 10.13 | – | 2.2E-12 |
| GOBP_NEGATIVE_REGULATION_OF_MEGAKARYOCYTE_DIFFERENTIATION | 20 | 17 | 85 | 8.20 | – | 4.7E-02 |
| GOCC_CYTOSOLIC_LARGE_RIBOSOMAL_SUBUNIT | 60 | 55 | 92 | 6.34 | – | 4.7E-07 |
| GOCC_CYTOSOLIC_SMALL_RIBOSOMAL_SUBUNIT | 41 | 36 | 88 | 4.00 | – | 3.1E-02 |
| GOCC_NUCLEOSOME | 134 | 97 | 72 | 3.78 | – | 4.8E-07 |
| GOCC_CYTOSOLIC_RIBOSOME | 118 | 107 | 91 | 3.50 | – | 4.8E-07 |
| GOBP_NUCLEOSOME_ORGANIZATION | 138 | 105 | 76 | 3.40 | – | 1.2E-06 |
| GOMF_STRUCTURAL_CONSTITUENT_OF_RIBOSOME | 169 | 153 | 91 | 3.00 | – | 1.4E-07 |
| GOCC_LARGE_RIBOSOMAL_SUBUNIT | 117 | 111 | 95 | 2.80 | – | 1.2E-04 |
| GOCC_RIBOSOMAL_SUBUNIT | 188 | 177 | 94 | 2.66 | – | 4.5E-07 |
| GOBP_RIBOSOMAL_LARGE_SUBUNIT_BIOGENESIS | 76 | 73 | 96 | 2.53 | – | 4.4E-02 |
| GOCC_CATALYTIC_STEP_2_SPLICEOSOME | 91 | 88 | 97 | 2.43 | – | 2.0E-02 |
| GOBP_CYTOPLASMIC_TRANSLATION | 156 | 146 | 94 | 2.41 | – | 1.7E-04 |
| GOCC_PRERIBOSOME | 109 | 105 | 96 | 2.18 | – | 3.6E-02 |
| GOCC_RIBOSOME | 239 | 215 | 90 | 2.17 | – | 2.6E-05 |
| GOBP_PROTEIN_DNA_COMPLEX_ASSEMBLY | 240 | 189 | 79 | 2.08 | – | 5.1E-04 |
| GOMF_STRUCTURAL_MOLECULE_ACTIVITY | 809 | 446 | 55 | 1.87 | – | 4.5E-07 |
| GOCC_RIBONUCLEOPROTEIN_COMPLEX | 1169 | 661 | 57 | 1.70 | – | 2.0E-07 |
| GOBP_RIBOSOME_BIOGENESIS | 325 | 308 | 95 | 1.67 | – | 6.2E-03 |
| GOBP_RIBONUCLEOPROTEIN_COMPLEX_BIOGENESIS | 502 | 447 | 89 | 1.62 | – | 5.6E-04 |
| GOCC_GOLGI_APPARATUS | 1634 | 1068 | 65 | 0.77 | 1.30 | 4.6E-02 |
| GOBP_LYMPHOCYTE_ACTIVATION | 796 | 405 | 51 | 0.65 | 1.53 | 4.4E-02 |
| GOMF_PROTEIN_KINASE_ACTIVITY | 577 | 382 | 66 | 0.60 | 1.66 | 6.2E-03 |
| GOBP_REGULATION_OF_ANATOMICAL_STRUCTURE_MORPHOGENESIS | 937 | 488 | 52 | 0.60 | 1.67 | 5.1E-04 |
| GOCC_BASEMENT_MEMBRANE | 90 | 49 | 54 | 0.15 | 6.46 | 4.4E-03 |

## Discussion

The present study compared gene expression data generated using TempO-Seq on cell lysates with RNA-Seq on purified RNA samples for 39 cell lines from many different types of tissue. The main goal of the study was to assess the comparability of gene expression measurements from the two technologies and compatibility for TempO-Seq and RNA-Seq whole transcriptome analysis to be combined for integrated analysis. To our knowledge, this study is the first of its kind; previous studies have all performed comparisons using only purified RNA for both platforms.

The first component of our analysis compared two TempO-seq data sets that had different read depths (6 million reads for Phase 1 and 4.5 million reads for Phase 2) for six different cell lines containing three technical replicates each and found that $\log_2$(CPM+1) normalized gene counts for all of the technical replicates from the same cell line were highly correlated

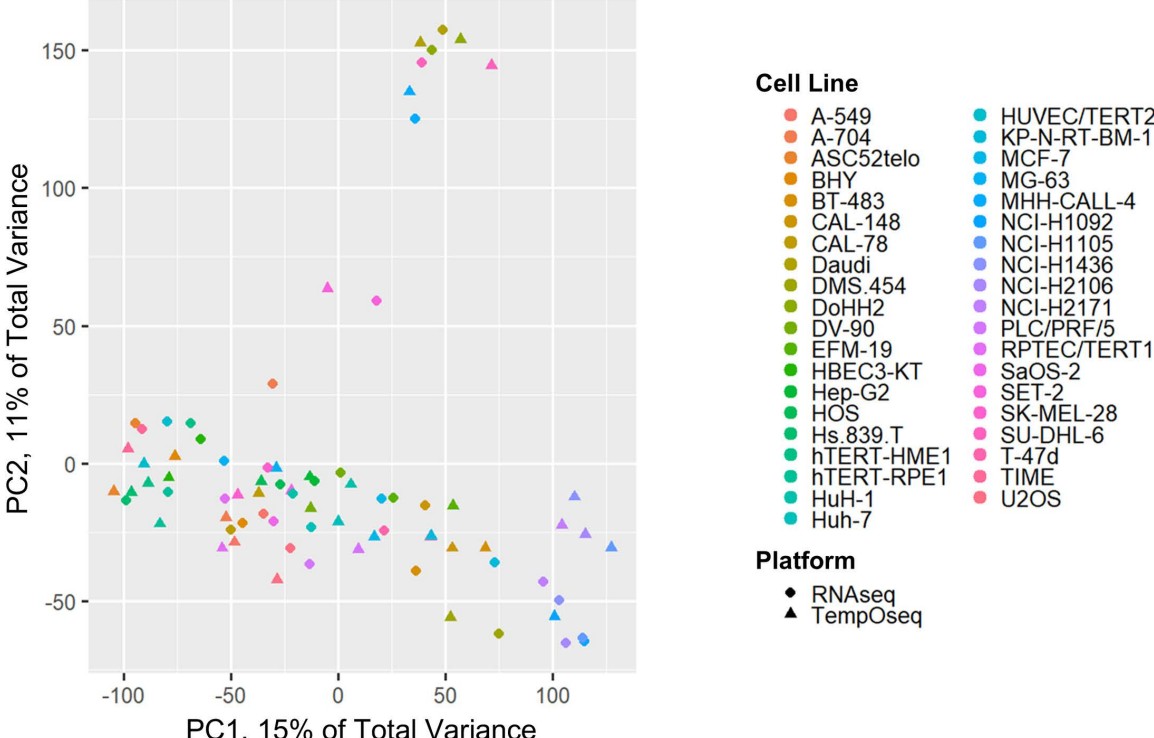

**Fig 6. TempO-seq vs RNA-seq principal component analysis (PCA) using relative log expression (RLE) normalized data.** RLE on 19,290 overlapping genes resolved the TempO-seq versus RNA-seq platform divergence observed within the PCA. The cell lines grouped together by cell line (color) instead of by technological platform (circles for RNA-seq and triangles for TempO-seq). The cell lines in the off-shoot with PC2 > 50 are all cancer cell lines derived from the immune system: SET-2, MHH-CALL-4, DoHH2, SU-DHL-6, and Daudi.

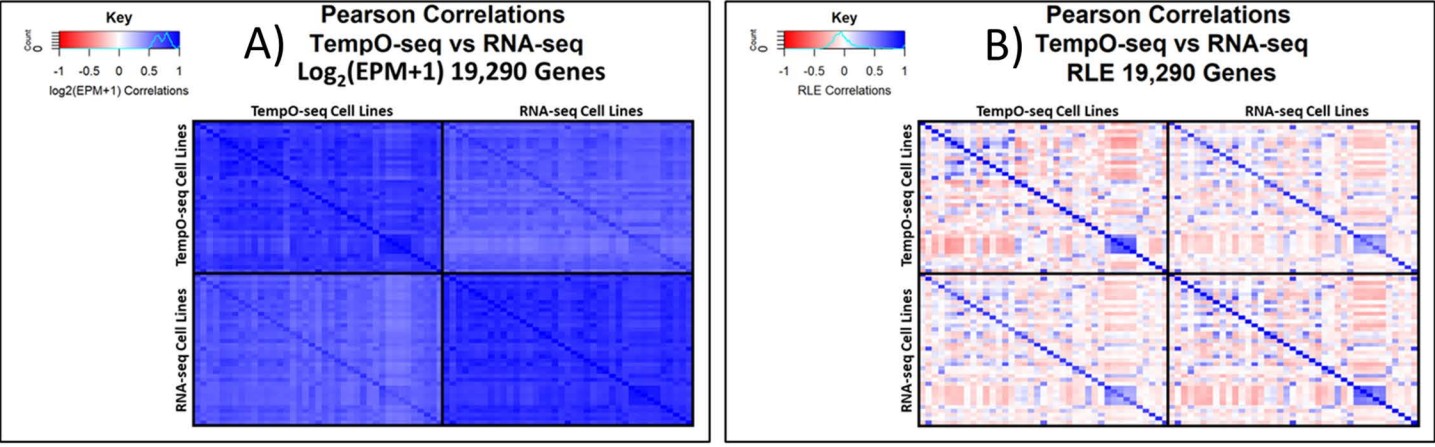

**Fig 7. Heatmaps of Pearson correlations for log₂ and RLE data.** Quadrants within each figure panel show Pearson correlations between the 39 cell lines within TempO-seq (top left quadrant), between the TempO-seq and RNA-seq cell lines (bottom left and top right quadrants), and between the 39 cell lines within RNA-seq (bottom right quadrant). The diagonal within each quadrant is the Pearson correlations between the same cell line. (A) Heatmap of Pearson correlations for the 19,290 overlapping genes with units of $\log_2$(expression per million + 1), abbreviated to $\log_2$(EPM+1). (B) Heatmap of Pearson correlations from the relative log expression (RLE) normalization approach. The full Pearson correlation numerical data used to generate the heatmaps is available in the supplemental S2 File.

within and across the two data sets. We further identified that the data was influenced much more strongly by cell line identity (78% of variance across all PCs) and not by whether the replicate was part of the Phase 1 or Phase 2 data set (< 10% of variance). Considering that the cell culture and lysates for Phase 1 and Phase 2 were produced months apart and at different read depths, this demonstrates a high degree of reproducibility for TempO-seq generated in the same laboratory over time (by the same laboratory technician from the same cryostocks of cells), even with mapped read depths that differed substantially.

The second component of our analysis compared the TempO-seq whole transcriptome data generated from cell lysates versus RNA-seq data generated from purified RNA across 39 cell lines. The $\log_2$(EPM+1) normalized gene count data was highly correlated despite the cultures and samples being prepared by different laboratories with different cell stocks. Of note, the HPA RNA-seq data was available only as an average of their three technical replicates, so we were unable to evaluate the technical replicability for the RNA-seq data like we were able to conduct for the TempO-seq data. In spite of this limitation, we determined that the median for the interquartile range for the relative log difference in $\log_2$(EPM + 1) was nearly equivalent, being centered near zero (-0.04, see Fig 3) and the Pearson correlations were strong (0.77, 95% CI: 0.76–0.78, see Fig 7). This demonstrates a high degree of concordance in gene expression profiles produced by TempO-seq and RNA-seq.

However, despite the high correlations in $\log_2$(EPM + 1) data between the two platforms, further statistical tests and data visualizations (PCA, PCoA, *betadisper*, and PERMANOVA) revealed a clear divergence between the TempO-seq and RNA-seq data. From this we identified that a subset of 20% of the genes had expression levels that differed substantially between platforms (Fig 5). Encouragingly, the percentage of genes that we found to be non-concordant between TempO-seq and RNA-seq (20%) was comparable and actually less than the 25% percent of genes that Angel and colleagues had found to be heavily influenced by technological platform when they combined 38 transcriptomics data sets for multiple types of human blood cells [31]. Additionally, the variance we observed between the TempO-seq and RNA-seq data was also similar to the differences noted by Yeakley et al in their comparison of the two technologies [4]. Therefore, the variance in expression levels that we observed between the two technologies is akin to what others have reported in the literature when comparing data from different labs and different transcriptomics platforms.

To better understand the TempO-seq versus RNA-seq platform divergence seen in the initial PCA across all PCs, we identified the genes that were the most non-concordant in their expression between TempO-seq and RNA-seq (in which minimally expressed genes with a larger relative $\log_2$ difference between TempO-seq and RNA-seq were removed until the platform effect on the PCA was < 10% of total variance). There was a similar number of genes with higher expression vs with lower expression in TempO-seq compared to RNA-seq. The full list of these 3,810 non-concordant genes is available in the supplemental S2 File within the "TR_Nonconcordant genes" tab. Of note, the genes that were flagged as non-concordant either had consistently high or low expression in TempO-seq relative to RNA-seq, regardless of cell line. The consistency of the discrepancy across all 39 cell lines implies that the observed differences in these non-concordant genes is likely due to actual differences in the technological platforms.

When focusing on the subset of genes that were non-concordant, histone and ribosomal gene families made up a large proportion of those genes with the largest differences in expression levels. There were 246 genes with a difference of >5 $\log_2$(EPM + 1), for which over one quarter (27%) of those genes (n=66) were ribosomal (n=39) or histone (n=27) genes. Also, for the 66 genes with a difference of >7 $\log_2$(EPM + 1), half of those genes (n=33) were ribosomal (n=20) or histone (n=13) genes. Of all of the genes in the histone family, 73% of them

were non-concordant. The difference in detection of histone gene family expression levels was expected though because histone genes do not have poly-A tails [32] and the RNA-seq preparation procedure included a poly-A tail pull-down step. Thus, RNA-seq reported mostly very low TPM expression values for histone genes. In contrast, TempO-seq does not require poly-A tail pull-down during library preparation steps [4], and the data had high CPM values for histone family genes. This means that TempO-seq would be preferable to RNA-seq library preparations employing poly-A enrichment when interpreting expression levels for histone genes.

On the other hand, for unclear reasons, more than half of the genes for ribosomal proteins were non-concordant, in which TempO-seq probes were frequently not as efficient at detecting mRNA for ribosomal proteins. Thus, RNA-seq may be a preferable option when studying ribosomal protein genes. One possible explanation is that the TempO-seq probe design for a subset of the ribosomal protein mRNA did not reliably capture expression for those specific genes.

When analyzing gene ontology signatures for their proportion of non-concordant vs concordant genes that met a minimum expression threshold ($\geq 5$ EPM), we found that there were several ontologies with OR > 1 that were enriched for non-concordant genes (Table 3). Every GO signature with OR > 1 contained many histone-related genes and/or many ribosomal protein genes (RPL, RPS, and mitochondrial ribosomal protein genes such as MRPL). As a result, all statistically significant GO signatures with poor concordance had functions that were primarily related to those gene families, including chromatin localization and ribosomal subunit constituents. GO signatures with OR < 1 were enriched for genes with more concordant expression between TempO-seq and RNA-seq. The significant GO signatures that contain a higher proportion of concordant genes (Table 3) had functions related to cellular structure and kinase activity, which is encouraging considering that those types of functions are vital to all cell types.

Overall, the expression levels for these non-concordant genes should be interpreted with caution until further studies are conducted to better understand the reason for the discrepancy in the observed expression levels across technologies. Until then, more confidence can be placed in genes that had concordant measurements for baseline expression level across both platforms for all 39 cell lines, especially for genes in this analysis that were expressed at baseline (see the supplemental S2 File, "TR_Overlapping genes_categories" tab).

After gaining this understanding that the non-concordant genes had consistently higher or lower expression within each technology, we determined that we could resolve the platform divergence by employing an RLE normalization strategy. To compute RLE for each gene, we calculated the ratio of the $\log_2(\text{EPM} + 1)$ for each cell line compared to the average $\log_2(\text{EPM} + 1)$ across all cell lines for the corresponding technological platform, which was successful at removing the platform divergence observed across all PCs in the PCA that used the RLE values. RLE normalization was able to resolve the divergence because it drastically reduced the technological platform differences and highlighted the differences in cell line expression patterns instead. Interestingly, after RLE normalization, the immune system cancer cell lines and liver cell lines became distinguishable in their own distinct clusters separate from other cell types in the PCA. RLE normalization also brought the NCI lung cancer cell lines more closely together in the PCA. This shows that RLE was an efficient method for helping to group together cell lines from the same tissue or disease state within PCA, and this demonstrates that cell line specific differences in gene expression patterns are likely what was driving the main variance in the data after RLE normalization.

A main advantage of the RLE normalization method is that it was able to resolve the TempO-seq vs RNA-seq platform divergence without removing any genes. Additionally,

even though the TempO-seq vs RNA-seq Pearson correlations for matching cell types were slightly decreased after RLE, the Pearson correlations for non-matching cell types dropped to around zero. This was advantageous for highlighting cell specific effects and thus being able to very clearly differentiate between different samples. The primary disadvantage of the RLE method is that the rank order of genes within a single sample based on RLE expression no longer reflects the absolute expression level of those genes. Thus, this normalization method should only be used when comparing relative expression levels between different samples (for example, comparing the expression of PPARG gene in sample 1 vs sample 2) and cannot be used for analyses that would include comparing expression levels of genes within a single sample (for example, comparing PPARG vs PPARA expression levels within a single sample). In summary, the RLE normalization approach was highly effective for resolving platform differences and could be utilized by researchers who wish to combine TempO-seq and RNA-seq data for detecting differences between sample groups.

### Strengths and limitations

Two main strengths of this study include being the first to run this technological comparison using TempO-Seq data from lysed cells (instead of purified RNA), and the wide breadth of biological diversity covered by the cell lines. The 39 cell lines we included in this analysis contained both cancerous and non-cancerous subtypes and were from 11 different types of tissue: lung, lymphatic (lymphoma), liver, kidney, breast, bone, eye, blood (leukemia), endothelium (microvascular), skin, adipose, and brain. This represents a significant expansion upon previous studies, which tested liver and breast primary cells, and 5 cell lines (MCF-7 cells, PC-3 cells, and HL-60 cells in 3 differentiation states), thus covering 4 tissue types: blood, breast, liver, and prostate cancer [4,10,17,18,31].

Another important aspect of our analysis is that the TempO-seq and RNA-seq data sets were generated by different laboratories (EPA and the HPA groups, respectively). Thus, some of the detected differences are likely due to differences in laboratory conditions and/or cellular stocks instead of from technological platform differences, and a limitation of our analysis is that we cannot conclusively distinguish between these. However, it is also an advantage to see the real-world replicability of these datasets when being generated by completely different labs with different cell stocks using different transcriptomic platforms. As opportunities arise, other laboratories with access to complementary TempO-seq and RNA-seq data sets could evaluate how similar their list of non-concordant genes is to the list we have generated. If there is consistency in the list of genes that have substantial relative log expression differences in TempO-seq vs RNA-seq data found in other studies too, then that would provide further evidence of the differences being due to technological platform effects and not because of laboratory, technician, and/or cell culture effects.

Another main limitation of this work is the inability to evaluate the 8,803 genes that were not expressed at baseline in these 39 cell lines and the inability to evaluate genes that become induced upon various chemical exposures or biological treatments/changes. Thus, further research is needed to compare data following exposures to exogenous agents (e.g., chemicals) that will induce expression of those genes and then repeat the analysis of TempO-seq data generated from cell lysates versus RNA-seq data generated from purified RNA. The statistics are also somewhat limited by having only one set of measurements per cell line per platform for the TempO-seq vs RNA-seq comparisons, which was due to the lack of replicate data being available for public download from HPA. However, we found that the expression level comparisons were very consistent across all 39 different cell lines, which shows a robustness of the findings.

## Conclusions

To our knowledge, this was the first analysis comparing human whole transcriptome TempO-seq data from cell lysates with RNA-seq data generated from purified RNAs and is the first to compare the technologies across such a large variety of cell lines with strong biological coverage of many different organs and tissue types. Based on our analysis, TempO-seq data generated months apart at different read depths were highly reproducible, and TempO-seq data was highly correlated with RNA-seq data, with over 80% of genes having concordant expression levels. We have supplied a list of genes that had non-concordant expression levels between the platforms (including many histone and ribosomal genes, see the supplemental S2 File) for which expression data comparisons may need to be interpreted more cautiously until further studies are conducted to replicate these results and to understand the reason(s) for their non-concordance. Overall, we found that the observed platform-driven differences in $\log_2$ expression were around the normal variation that is expected for biological studies conducted in independent labs using different cell stocks. Even though there was a clear platform divergence effect within the initial $\log_2$ PCA between the TempO-seq and RNA-seq data, this divergence was readily resolved using RLE normalization (see methods for equation). RLE revealed similarity in gene expression profiles of cell lines from the same tissue and/or disease state in addition to smoothing out technological platform variance (as evidenced by the PCA and PCoA plots). Use of RLE could enable comparisons across other TempO-seq and RNA-seq data sets and could enable the possible combination of data from these platforms for a wide variety of data mining applications.

In conclusion, TempO-seq data using lysed cells was highly reproducible at different read depths and was highly concordant with RNA-seq data for most genes across 39 cell lines under baseline conditions, especially after RLE normalization, making it a comparable alternative technology for targeted transcriptomics studies.

## Supporting information

**S1 File.  Cell line information and gene expression data used for the analysis.** This file contains additional TempO-seq and RNA-seq information and data for all 39 cell lines included in the study, including but not limited to: the cell culture conditions, the TempO-seq probes and gene expression data for all replicates used for the TempO-seq Phase 1 vs Phase 2 comparison using 6 cell lines, and the gene expression and relative log expression (RLE) data for the TempO-seq vs RNA-seq comparison for all 39 cell lines.
(XLSX)

**S2 File.  Results of the analyses comparing TempO-seq Phase 1 vs Phase 2 and TempO-seq vs RNA-seq.** This file includes PCA plots and PERMANOVA statistical findings for the TempO-seq Phase 1 vs Phase 2 comparison using 6 cell lines and the Pearson correlations, PCA plots, and PERMANOVA statistical findings for the log2 and relative log expression (RLE) TempO-seq vs RNA-seq comparisons using 39 cell lines. This file also includes the full list of genes within each gene ontology (GO) signature and the GO statistical p-values and false discovery rate (FDR) results, as well as the complete list of 3,810 non-concordant genes between TempO-seq and RNA-seq.
(XLSX)

**S3 File.  Principal coordinates analysis (PCoA) and dispersion statistics for TempO-seq Phase 1 vs 2 and TempO-seq vs RNA-seq.** This file includes the PCoA and dispersion statistical findings for the TempO-seq Phase 1 vs Phase 2 comparison using 6 cell lines, and the

PCoA and dispersion statistical findings for the TempO-seq vs RNA-seq comparison using 39 cell lines.
(PDF)

## Acknowledgments

We would like to acknowledge Drs. Leah Wehmas, Brian Chorley, Katie Paul Friedman, and E. Sidney Hunter for their critical technical review of this manuscript.

## Author contributions

**Conceptualization:** Laura J. Word.

**Data curation:** Laura J. Word, Clinton M. Willis, Logan J. Everett, Derik E. Haggard, Joshua A. Harrill.

**Formal analysis:** Laura J. Word.

**Funding acquisition:** Nisha S. Sipes, Joshua A. Harrill.

**Investigation:** Laura J. Word, Clinton M. Willis.

**Methodology:** Laura J. Word, Richard S. Judson, Logan J. Everett, Sarah E. Davidson-Fritz, Bryant A. Chambers, Joseph L. Bundy, Imran Shah, Nisha S. Sipes, Joshua A. Harrill.

**Resources:** Clinton M. Willis, Joshua A. Harrill.

**Software:** Laura J. Word, Bryant A. Chambers, Jesse D. Rogers, Joseph L. Bundy.

**Supervision:** Richard S. Judson, Joshua A. Harrill.

**Validation:** Laura J. Word.

**Visualization:** Laura J. Word.

**Writing – original draft:** Laura J. Word.

**Writing – review & editing:** Laura J. Word, Clinton M. Willis, Logan J. Everett, Sarah E. Davidson-Fritz, Derik E. Haggard, Jesse D. Rogers, Joseph L. Bundy, Imran Shah, Nisha S. Sipes, Joshua A. Harrill.

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
