## [Decision Letter · Decision Letter 0]

29 Dec 2024

PONE-D-24-50333TempO-seq and RNA-seq Gene Expression Levels are Highly Correlated for Most Genes: A Comparison Using 39 Human Cell LinesPLOS ONE

Dear Dr. Word,

Thank you for submitting your manuscript to PLOS ONE. After careful consideration, we feel that it has merit but does not fully meet PLOS ONE’s publication criteria as it currently stands. Therefore, we invite you to submit a revised version of the manuscript that addresses the points raised during the review process.

We look forward to receiving your revised manuscript.

Kind regards,

Mingli Li

Academic Editor

PLOS ONE

3. We noted in your submission details that a portion of your manuscript may have been presented or published elsewhere. [Publicly available RNA-seq data from the Human Protein Atlas (HPA) was included in this analysis in combination with our own data from TempO-seq. The HPA data is open-access for researchers to use.] Please clarify whether this [conference proceeding or publication] was peer-reviewed and formally published. If this work was previously peer-reviewed and published, in the cover letter please provide the reason that this work does not constitute dual publication and should be included in the current manuscript.

Reviewers' comments:

Reviewer's Responses to Questions

**Comments to the Author**

1. Is the manuscript technically sound, and do the data support the conclusions?

Reviewer #1: Yes

Reviewer #2: Yes

2. Has the statistical analysis been performed appropriately and rigorously? 

Reviewer #1: I Don't Know

Reviewer #2: Yes

3. Have the authors made all data underlying the findings in their manuscript fully available?

Reviewer #1: Yes

Reviewer #2: Yes

4. Is the manuscript presented in an intelligible fashion and written in standard English?

Reviewer #1: Yes

Reviewer #2: Yes

5. Review Comments to the Author

Reviewer #1: A straight-forward comparison of the TempO-Seq data obtained from cell lysates and published RNA seq data for the same cell line in the absence of any exposure (basal expression). The authors found that there is significant correlation between the observed relative expression level as determined by TempO-seq and the relative expression level determined by RNA seq of Poly A purified mRNA for the majority of transcripts. The authors observed discordance for a significant portion of transcripts (3,810 out of 19,290) between the two platforms which they further investigated through enrichment analysis of the discordant transcripts . Histone transcripts without polyA tails showed significantly lower expression levels in RNA seq data as expected. They also noted enrichment of ribosomal protein genes in the discordant transcripts with underrepresentation in the TempO seq data for unknown reasons. The authors suggest that comparative analysis of these 3810 transcripts should be considered with caution due to the clear platform differences. Of the remaining transcripts, they demonstrate and demonstrate that Relative Log Expression (RLE) normalization of the data results in high concordance between platforms indicating that direct comparisons between platforms using this approach for the concordant transcripts is reasonable.

Minor issues

Line 29 – the meaning of overlapping is unclear – perhaps reword

Line 31 – The statement that RLE alone resolved platform divergence may not be exactly correct as I believe that they have to remove the discordant set of transcripts from the comparison initially.

Line 68 - the statement that TempO-seq does not generate much information about transcripts variants is confusing – as far as I understand, the approach is dependent on the probes used so it could provide no data on variants or could detect all variants if designed to do so – consider rewording.

Line 76 – the authors use “This is…” repeatedly throughout the manuscript and the reader is often left to guess to what the author is referring. Please consider specifying exactly to what the authors are referring to remove ambiguity.

Line 86 - reword “gene expression as compared to RNA-seq”

Line 199 – Readers may not be familiar with principal coordinate analysis as compared to principal component analysis and the authors may want to consider a brief explanation of the differences between the two and their use in this context.

Line 416 – the sentence beginning “The average…” is confusing, please reword.

Line 522 – RLE is defined only in the methods before its mention here in the results – the authors may want to consider a brief explanation here for clarity for the reader.

Line 631 – Are there any features of ribosomal mRNA sequence or processing that would lead to increased likelihood of poor probe design as compared to other mRNA?

Line 680- technological is misspelled

Reviewer #2: This manuscript presents a comprehensive comparison of TempO-seq and RNA-seq technologies in profiling gene expression across 39 human cell lines. First, the authors validated the consistency of TempO-seq at different read depths, and then performed a direct comparison between TempO-seq and RNA-seq. Additionally, the authors incorporated the method to mitigate batch effects between the two technologies. This study provides valuable insights into the comparability of TempO-seq and RNA-seq for transcriptomics research.

Does TempO-seq also capture other types of RNA beyond mRNA, such as lncRNA? Standard RNA-seq libraries typically cover lncRNA as well. TempO-seq appears to be a targeted sequencing method, which may limit its effectiveness in screening all expressed features across the genome. Do the authors have any comments on this? Would this be considered a limitation of the method?

The manuscript would also benefit from discussing the price and turnaround time for TempO-seq compared to RNA-seq. This information is critical when deciding which method to use for specific applications.

Regarding the selection of the 39 human-derived cell lines for comparison, why did the authors focus on these particular cell lines, most of which are derived from cancer tissues? Can the findings be extrapolated to normal tissues? Would it be beneficial to include a comparison using mouse samples to further validate the results?

In Lines 360 and 384, how did the authors quantify the variance attributable to the TempO-seq phase designation?

6. PLOS authors have the option to publish the peer review history of their article (what does this mean? ). If published, this will include your full peer review and any attached files.

**Do you want your identity to be public for this peer review?** For information about this choice, including consent withdrawal, please see our Privacy Policy .

Reviewer #1: No

Reviewer #2: No

---

## [Author Response · Author response to Decision Letter 1]

11 Feb 2025

Responses to Reviewer 1)

Reviewer #1: A straight-forward comparison of the TempO-Seq data obtained from cell lysates and published RNA seq data for the same cell line in the absence of any exposure (basal expression). The authors found that there is significant correlation between the observed relative expression level as determined by TempO-seq and the relative expression level determined by RNA seq of Poly A purified mRNA for the majority of transcripts. The authors observed discordance for a significant portion of transcripts (3,810 out of 19,290) between the two platforms which they further investigated through enrichment analysis of the discordant transcripts . Histone transcripts without polyA tails showed significantly lower expression levels in RNA seq data as expected. They also noted enrichment of ribosomal protein genes in the discordant transcripts with underrepresentation in the TempO seq data for unknown reasons. The authors suggest that comparative analysis of these 3810 transcripts should be considered with caution due to the clear platform differences. Of the remaining transcripts, they demonstrate that Relative Log Expression (RLE) normalization of the data results in high concordance between platforms indicating that direct comparisons between platforms using this approach for the concordant transcripts is reasonable.

We appreciate all of your helpful feedback. We have addressed your concerns, as detailed below. (Note that the line numbers are slightly different since they correspond to the newly edited document with tracked changes).

Minor issues

*Line 29 – the meaning of overlapping is unclear – perhaps reword

We re-worded the sentence:

Line 31: “The log2 normalized expression data for 19,290 genes within both platforms were well correlated between TempO-seq and RNA-seq (Pearson correlation 0.77, 95% CI: 0.76 – 0.78), and the majority of genes (15,480 genes, 80%) had concordant gene expression levels.”

*Line 31 – The statement that RLE alone resolved platform divergence may not be exactly correct as I believe that they have to remove the discordant set of transcripts from the comparison initially.

The RLE analysis included all genes. This point has been clarified in the methods:

Line 331-333: “We did this for each gene and cell line by dividing each gene’s log2(EPM + 1) by the average log2(EPM + 1) across all cell lines for the TempO-seq data and the RNA-seq data separately for all 19,290 genes.”

To elaborate, performing the RLE transformation on all genes helped to preserve the relative differences in expression across different cell types while smoothing out the platform differences. For example, if a gene had high expression data across nearly all of the cell lines, but one cell line had much lower expression than the other cell lines, then most cell lines would have an RLE of around 1 except the cell line with lower expression which would therefore have an RLE < 1. Looking at the same gene in the other platform, this pattern would hold within the RLE data even if the raw data had comparatively lower expression levels. In the same scenario, most cell lines would have an RLE around 1, except the cell line with lower expression than the others, giving it an RLE < 1. In this way, RLE is highly effective at smoothing out the platform differences and highlighting the unique expression patterns within different cell lines. Thus, we were able to include all 19,290 genes in the RLE normalization and actually resolve the platform divergence observed within our various statistical tests (PCA, PCoA, dispersion tests).

*Line 68 - the statement that TempO-seq does not generate much information about transcripts variants is confusing – as far as I understand, the approach is dependent on the probes used so it could provide no data on variants or could detect all variants if designed to do so – consider rewording.

Edited to soften the phrasing as follows:

Line 70-72: “A potential disadvantage is that TempO-seq is not able to detect novel mRNA and may not generate much information about transcript variants, depending on the probes used.”

*Line 76 – the authors use “This is…” repeatedly throughout the manuscript and the reader is often left to guess to what the author is referring. Please consider specifying exactly to what the authors are referring to remove ambiguity.

The text through the manuscript has been edited slightly in several places to make the meaning of "This is..." clear in each case.

Line 79-81: This use of non-functional competitor DOs is beneficial because sequencing reads are diverted to less abundantly expressed genes without the added expense of increasing read depth.

Line 666-668: “RLE normalization was able to resolve the divergence because it drastically reduced the technological platform differences and highlighted the differences in cell line expression patterns instead."

Line 705-709: "However, it is also an advantage to see the real-world replicability of these datasets when being generated by completely different labs with different cell stocks using different transcriptomic platforms."

*Line 86 - reword “gene expression as compared to RNA-seq”

We reworded this sentence to:

Line 89-90: “Limited case studies have demonstrated that TempO-seq is as consistent and sensitive at detecting changes in gene expression as RNA-seq.”

*Line 199 – Readers may not be familiar with principal coordinate analysis as compared to principal component analysis and the authors may want to consider a brief explanation of the differences between the two and their use in this context.

We have added the following elaboration about the key difference between the two techniques:

Line 207-209: “While PCA reduces data to a smaller number of key components that capture the most variation (PCs), PCoA is a technique that visualizes how different samples relate to one another based on their overall dissimilarity, often using distance measures (23).”

*Line 416 – the sentence beginning “The average…” is confusing, please reword.

We have reworded the sentence as follows:

Line 424-428: “The average 50th percentile for of the TempO-seq minus RNA-seq relative log difference in log2(EPM + 1) across all cell lines was -0.04, which indicates that there was only a slight difference on average between TempO-seq and RNA-seq expression values."

*Line 522 – RLE is defined only in the methods before its mention here in the results – the authors may want to consider a brief explanation here for clarity for the reader.

We added the meaning of RLE in the results section for reader’s benefit:

Line 530-533: “ANOVA on PC1 RLE data showed that the platform accounted for none of the variance following this relative log expression normalization step (R^2 = 0, p = 1).“

*Line 631 – Are there any features of ribosomal mRNA sequence or processing that would lead to increased likelihood of poor probe design as compared to other mRNA?

We are unaware of any features of ribosomal mRNA sequences or processing that would contribute to suboptimal probe designs compared to other mRNA; the probe design process used by the TempO-Seq manufacturers is proprietary, so we do not know the reason why they did not perform as expected.

*Line 680- technological is misspelled

We have corrected this spelling mistake (see its new line, 693), we appreciate you pointing it out to us.

Responses to Reviewer 2)

Reviewer #2: This manuscript presents a comprehensive comparison of TempO-seq and RNA-seq technologies in profiling gene expression across 39 human cell lines. First, the authors validated the consistency of TempO-seq at different read depths, and then performed a direct comparison between TempO-seq and RNA-seq. Additionally, the authors incorporated the method to mitigate batch effects between the two technologies. This study provides valuable insights into the comparability of TempO-seq and RNA-seq for transcriptomics research.

Thank you for your thoughtful review of our manuscript. We have responded to your questions below:

*Does TempO-seq also capture other types of RNA beyond mRNA, such as lncRNA? Standard RNA-seq libraries typically cover lncRNA as well. TempO-seq appears to be a targeted sequencing method, which may limit its effectiveness in screening all expressed features across the genome. Do the authors have any comments on this? Would this be considered a limitation of the method?

We agree that the standard TempO-Seq whole transcriptome assay as currently designed is not intended to measure other types of RNA besides mRNA, like lncRNAs, which could be considered a limitation of the technology in some circumstances. However, TempO-Seq is not limited to mRNAs. Sensitive detection of miRNAs or other ncRNAs is also possible and they can generate custom oligos upon request (BioSpyder).

*The manuscript would also benefit from discussing the price and turnaround time for TempO-seq compared to RNA-seq. This information is critical when deciding which method to use for specific applications.

We recognize that pricing and turnaround time are important considerations in the pros vs cons when selecting which platform to use for running transcriptomics data. However, the pricing structure for TempO-seq is too variable for us to be able to address within the manuscript; TempO-seq pricing depends on the details of the contract established with Biospyder if a laboratory is using them as a service provider, or whether a researcher purchases TempO-Seq kits from BioSpyder to run with their own laboratories or facilities.

With regard to turn-around time, this will also be variable between labs, but the introduction does highlight that the sample preparation and data cleaning for RNA-seq are time-consuming and resource intensive:

Lines 52-57: “RNA-seq involves fragmentation of RNA, reverse transcription, amplification, and subsequent sequencing and alignment of reads to a reference transcriptome (which requires significant computing resources) (3). There are different library preparation methods for RNA-seq, but a common approach enriches for messenger RNA (mRNA) through a poly-adenylated (poly-A) tail pull-down step (3), which is labor intensive and time-consuming.”

We also made it more clear that the TempO-seq sample and data preparation take less time:

Lines 61-64: “TempO-seq has been shown to be highly reproducible (5), and it has technical advantages of being compatible with cell lysates, thus saving time by eliminating the need for RNA purification and reverse transcription (4). TempO-seq also has a simpler and more straightforward alignment process of sequencing data to the TempO-seq probe manifest.”

*Regarding the selection of the 39 human-derived cell lines for comparison, why did the authors focus on these particular cell lines, most of which are derived from cancer tissues? Can the findings be extrapolated to normal tissues? Would it be beneficial to include a comparison using mouse samples to further validate the results?

The cell lines used in this study were selected based on results from a separate project aimed at selecting cell lines for in vitro high-throughput transcriptomics screening.

When comparing the cancer cell lines and hTERT immortalized normal human cell lines, we found it reassuring that they had consistent trends when looking at the relative log expression differences between TempO-seq and RNA-seq data.

One of the main reasons for us to conduct this study was to substantially expand the analysis of TempO-seq vs RNA-seq across a much greater human biological space than has been previously examined, especially since there are prior studies that have already looked further into rodent data (Rats: Bushel et al 2018, Mice: Cannizzo et al 2022). Those studies found similar results as our study (i.e., that TempO-seq and RNA-seq are comparable), so we feel that question has already been sufficiently addressed in the literature.

The introduction already detailed that Bushel et al studied rat samples, and we have added to the manuscript the detail that Cannizzo et al’s study was a comparison of the technologies using mouse liver samples:

Line 114-117: “More recently, Cannizzo and colleagues determined that TempO-seq provided more consistent fold-change results for differentially expressed genes (DEGs) within frozen and FFPE mouse liver samples whereas RNA-seq had much greater variation between the two sample types (10).”

*In Lines 360 and 384, how did the authors quantify the variance attributable to the TempO-seq phase designation?

R^2 is indicative of the percent variance explained for the variable being evaluated; the variance within PC1 that is attributable to the TempO-seq phase designation came directly from the R^2 value from the ANOVA results run on PC1 data, converted into a percentage.

---

## [Editor Report · Decision Letter 1]

26 Feb 2025

TempO-seq and RNA-seq Gene Expression Levels are Highly Correlated for Most Genes: A Comparison Using 39 Human Cell Lines

PONE-D-24-50333R1

Dear Dr. Word,

We’re pleased to inform you that your manuscript has been judged scientifically suitable for publication and will be formally accepted for publication once it meets all outstanding technical requirements.

Kind regards,

Mingli Li

Academic Editor

PLOS ONE

---

## [Editor Report · Acceptance letter]

PONE-D-24-50333R1

PLOS ONE

Dear Dr. Word,

I'm pleased to inform you that your manuscript has been deemed suitable for publication in PLOS ONE. Congratulations! Your manuscript is now being handed over to our production team.

Kind regards,

on behalf of

Dr. Mingli Li

Academic Editor

PLOS ONE